# DddA homolog search and engineering expand sequence compatibility of mitochondrial base editing

Li Mi[1,9], Ming Shi[1,2,9], Yu-Xuan Li[1], Gang Xie [3], Xichen Rao [4,5], Damu Wu[4], Aimin Cheng[2], Mengxiao Niu[1], Fengli Xu[1], Ying Yu[4,5,6,7,8], Ning Gao [2,4], Wensheng Wei [4,5,6,7,8], Xianhua Wang[1] & Yangming Wang [1] ✉

Expanding mitochondrial base editing tools with broad sequence compatibility is of high need for both research and therapeutic purposes. In this study, we identify a DddA homolog from *Simiaoa sunii* (Ddd_Ss) which can efficiently deaminate cytosine in D$\underline{C}$ context in double-stranded DNA (dsDNA). We successfully develop Ddd_Ss-derived cytosine base editors (DdCBE_Ss) and introduce mutations at multiple mitochondrial DNA (mtDNA) loci including disease-associated mtDNA mutations in previously inaccessible G$\underline{C}$ context. Finally, by introducing a single amino acid substitution from Ddd_Ss, we successfully improve the activity and sequence compatibility of DdCBE derived from DddA of *Burkholderia cenocepacia* (DdCBE_Bc). Our study expands mtDNA editing tool boxes and provides resources for further screening and engineering dsDNA base editors for biological and therapeutic applications.

Base editing enables precise mutations of a target DNA sequence and is useful for dissecting the function of genes and regulatory elements, disease modeling, and developing therapeutics to treat diseases[1]. Base editing in nuclear DNA has been achieved through clustered regularly interspaced short palindromic repeats (CRISPR) derived base editors[2–4]. Unfortunately, CRISPR base editors are currently not applicable for mtDNA editing due to the lack of approaches delivering guide RNAs into mitochondria[5]. Recently, transcription activator-like-effector (TALE) derived base editors DdCBEs[6,7] and TALEDs[8] have been developed to catalyze C-to-T and A-to-G editing in mtDNA. These methods depend on DddA$_{tox}$, a dsDNA deaminase from the bacterium *Burkholderia cenocepacia* (Ddd_Bc). The original Ddd_Bc requires a strict T$\underline{C}$ sequence context[6]. Through phage-assisted evolution, Mok *et al*. have further obtained a Ddd_Bc variant that can catalyze C-to-T editing in H$\underline{C}$ sequence context[7]. Moreover, de Moraes et al. have looked into

additional bacterial deaminase toxin families and identified deaminases with different sequence contexts and substrates (e.g., single- versus double-strand DNA) preferences[9]. However, DdCBEs suitable for G$\underline{C}$ context targets are still unavailable. In addition, whether more dsDNA deaminases can be found in other species is unknown.

In this study, we identify and engineer dsDNA deaminase homologs to address the above problems. Through PSI-BLAST and deamination assays, we discover a dsDNA deaminase from *Simiaoa sunii* (Ddd_Ss) that show high activity and broad sequence compatibility. Based on this dsDNA deaminase, we develop highly efficient base editing tools that introduce mutations at multiple mtDNA loci including disease-related mutations in previously inaccessible G$\underline{C}$ context. Finally, by introducing a single amino acid substitution from Ddd_Ss to Ddd_Bc, we successfully improve the activity and sequence compatibility of DdCBE_Bc.

[1]Institute of Molecular Medicine, College of Future Technology, Peking University, Beijing, China. [2]Peking-Tsinghua Center for Life Sciences, Peking University, Beijing, China. [3]Academy for Advanced Interdisciplinary Studies, Peking University, Beijing, China. [4]School of Life Sciences, Peking University, Beijing, China. [5]State Key Laboratory of Protein and Plant Gene Research, Peking University, Beijing, China. [6]Biomedical Pioneering Innovation Center, Peking University, Beijing, China. [7]Beijing Advanced Innovation Center for Genomics, Peking University, Beijing, China. [8]Peking University Genome Editing Research Center, Peking University, Beijing, China. [9]These authors contributed equally: Li Mi, Ming Shi. ✉e-mail: yangming.wang@pku.edu.cn

## Results

### SPKK-related motif is important for the deamination activity of Ddd_Bc

From amino acid sequence analysis of Ddd_Bc, we noticed that its C-terminus contains two SPKK-related peptide motifs[10] which are known to prefer the binding of A/T-rich DNA sequences in the minor groove of the dsDNA (Fig. 1a). Deletion of the two SPKK-related motifs completely abolished the dsDNA deaminase activity of Ddd_Bc (Fig. 1b, c and Supplementary Fig. 1a). Moreover, the addition of AT-hook[11] that has a similar DNA binding property as SPKK-related motifs restored the deaminase activity of truncated Ddd_Bc (Fig. 1b, c and Supplementary Fig. 1a). Previous studies indicate that proline-dependent β-turn structure is important for fitting SPKK peptide into the minor groove

of dsDNA[12]. Interestingly, mutation of prolines to valine or asparagine in C-terminal peptide reduced or completely abolished the deaminase activity of Ddd_Bc (Supplementary Fig. 1b, c). These data suggest that the SPKK-related motif at the C-terminus of Ddd_Bc is important for its dsDNA deamination activity, possibly through facilitating DNA binding or other unknown structural roles.

### Identification of homologs of Ddd_Bc with dsDNA deaminase activity

Next, we used PSI-BLAST[13] to identify homologs of Ddd_Bc. We successfully identified 555 candidate homologs of Ddd_Bc from the non-redundant protein database (nr50_1_Nov, 2021). We then picked eight candidates from the list (Supplementary Data 1), four with SPKK-

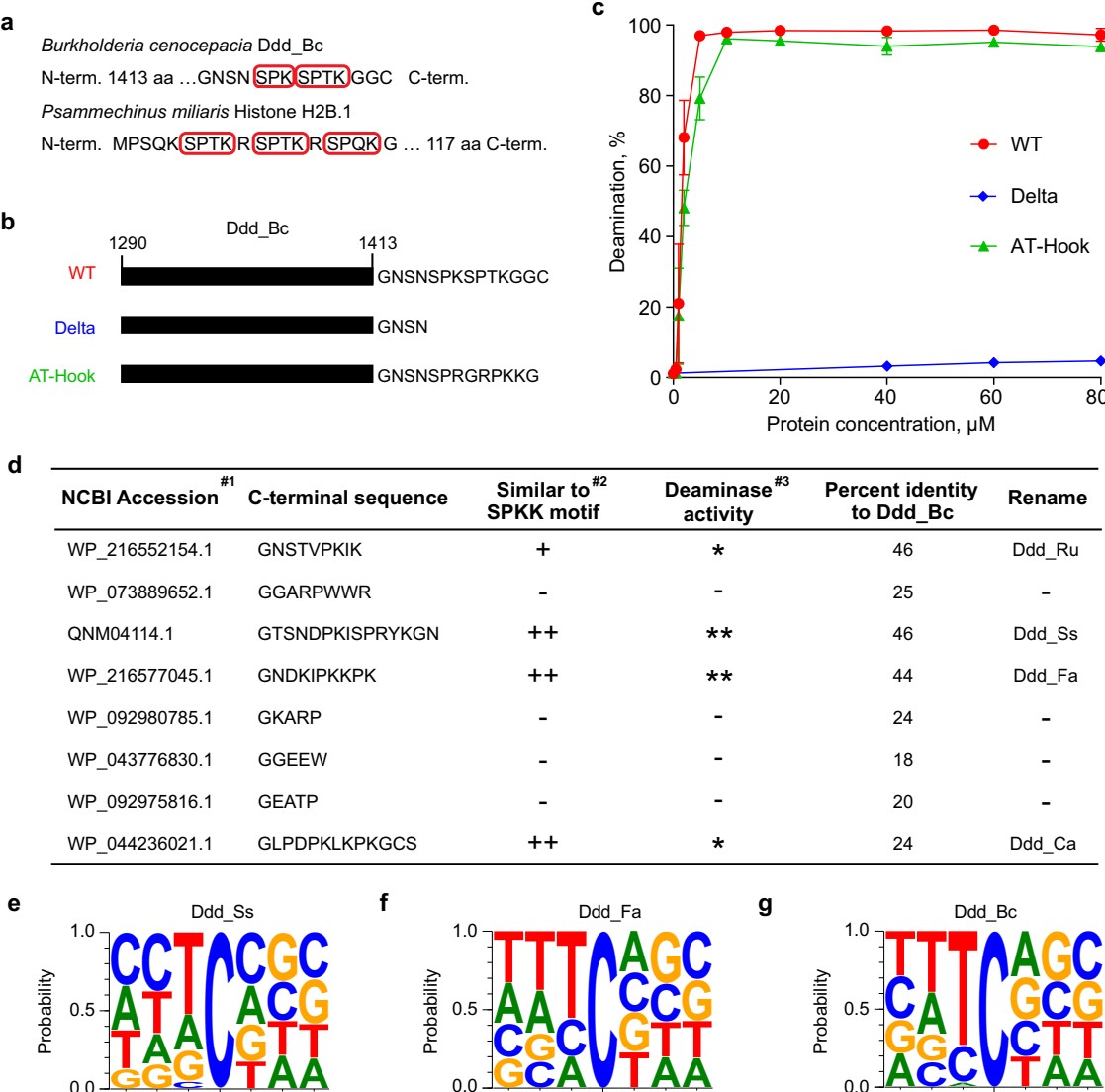

**Fig. 1 | SPKK-related motifs help identify Ddd_Bc homologs with dsDNA deaminase activity. a** The red boxes mark the SPKK-related motif sequences in *Burkholderia cenocepacia* Ddd_Bc and *Psammechinus miliaris* Histone H2B.1. **b** The schematic of constructs for wild type Ddd_Bc (WT), Ddd_Bc with two SPKK-related motifs removed (Delta), and Delta rescued with AT-hook sequence (AT-Hook). **c** Quantification of the relative amounts of deamination product versus protein concentration from different protein constructs as shown in **b**. Associate gels are shown in Supplementary Fig. 1a. Shown are mean ± SD; *n* = 3 independent

experiments. **d** Summary for the sequence features and the activity of candidate Ddd_Bc homologs. #1: Ordered by PSI-BLAST score, from high to low. #2: +, the C-terminal region has one SPKK-related motif; ++, two SPKK-related motifs; -, no SPKK-related motif. #3: *, activity comparable to Ddd_Bc; **, activity higher than Ddd_Bc; -, activity low or weak. Associated gels and quantifying results are shown in Supplementary Fig. 2a–j. **e**–**g** Probability sequence logo of the region flanking mutated cytosines from the genome of UNG-deficient *E.coli* overexpressing Ddd_Ss (**e**), Ddd_Fa (**f**), and Ddd_Bc (**g**). Source data are provided as a Source Data file.

related motifs and 4 without (Fig. 1d), to test their deaminase activity on dsDNA substrates. Interestingly, all four proteins containing SPKK-related motifs showed relatively high deamination activity, comparable to or higher than that of Ddd_Bc (Fig. 1d and Supplementary Fig. 2a–j). In contrast, all four proteins without SPKK-related motifs showed only a little or no deamination activity. These results suggest that SPKK-related motifs may be useful for the identification of dsDNA deaminases with high activities.

## Ddd_Ss is a dsDNA deaminase with high activity and broad sequence compatibility

In the deamination assay, we noticed multiple bands formed by a dsDNA deaminase from *Simiaoa sunii* (Ddd_Ss), indicating a broad deaminase activity at the non-TC context (Supplementary Fig. 2d). Indeed, deamination assay using four DNA substrates with different NC sequence contexts confirmed that Ddd_Ss could target cytosine residues preceded by A, G, C, or T, while Ddd_Bc only worked efficiently for TC context (Supplementary Fig. 2k). Furthermore, whole-genome sequencing of UNG-deficient *E. coli* strain expressing Ddd_Ss revealed that it has a preference for deamination at DC context (Fig. 1e), while another dsDNA deaminase Ddd_Fa had a sequence compatibility for HC with a slightly higher preference for TC (Fig. 1f), which is similar to DddA11 (renamed as Ddd_Bc11), an evolved variant of Ddd_Bc[7]. As expected, the original Ddd_Bc had a high preference for TC context[9] (Fig. 1g). Since Ddd_Ss complements Ddd_Bc and its derivative variants for deamination at GC context, and it showed the highest deamination activity among all the deaminases tested including Ddd_Bc (Supplementary Fig. 2j), we then focused on Ddd_Ss throughout this study.

## Development of DdCBE_Ss as a mtDNA base editor

We next tested Ddd_Ss in targeted mtDNA editing by fusing Ddd_Ss halves to TALE array proteins containing mitochondrial localization sequences (MTS). Based on the alignment of the predicted structure of Ddd_Ss by ColabFold[14] and the crystal structure of Ddd_Bc (Fig. 2a), the

N29 and N94 in Ddd_Ss corresponded to optimal splitting sites G1333 and G1397 reported for Ddd_Bc[6]. We designed the following mitochondrial DdCBE containing Ddd_Ss (DdCBE_Ss) according to the original DdCBE study[6]: a pair of mito-TALE-split-Ddd_Ss that contains an MTS, a TALE array, a Ddd_Ss half from the N29 or N94 split, and a UGI protein (Fig. 2b). C- and N-terminal halves of Ddd_Ss were linked to the right and left TALE, respectively. We first tested DdCBE_Ss targeting *MT-ND5*, a gene encoding NADH dehydrogenase 5 subunit of complex I (Supplementary Data 2). We found that DdCBE_Ss enabled C-to-T editing at multiple C sites and displayed different sequence preferences over DdCBEs containing Ddd_Bc (DdCBE_Bc) in the same orientation (Fig. 2c). Importantly, DdCBE_Ss with the splitting site at N94 (DdCBE_Ss_N94) achieved around 40% editing of C6 at GC context, while the editing efficiency for the C6 by DdCBE_Bc with the splitting site at G1397 (DdCBE_Bc_G1397) was only around 8% (Fig. 2c and Supplementary Fig. 3a, b). In addition, for C7 at GC context, the editing efficiency by DdCBE_Ss_N94 was around 33-fold higher than that by DdCBE_Bc_G1397 (Fig. 2c). For this *MT-ND5.1* site, the DdCBE_Ss_N29 constructs caused generally lower editing at every cytosine than DdCBE_Ss_N94, although its editing efficiency for two GC sites (C6, 6.1% and C7, 5.2%) was still much higher than DdCBE_Bc_G1333 (both <0.3%) (Fig. 2c and Supplementary Fig. 3a, b). These results support that we have successfully designed a DdCBE from Ddd_Ss to install mtDNA editing with an advantage over Ddd_Bc for GC context.

To test the generality of DdCBE_Ss in mtDNA editing, we then constructed DdCBE_Ss targeting *MT-ATP6*. We observed the highest editing efficiency around 38% and 24% for DdCBE_Ss_N94 and DdCBE_Ss_N29 (Fig. 2d and Supplementary Fig. 3c, d), respectively. In the remaining study, we used the G1397 split and N94 split for Ddd_Bc and Ddd_Ss, respectively. Encouraged by these results, we then tested editing for 8 additional sites in 6 mitochondrial genes (Supplementary Data 2) in HEK293T cells, DdCBE_Ss resulted in ~6–51% editing at these sites (Supplementary Figs. 4, 5). Furthermore, we found that DdCBE_Ss

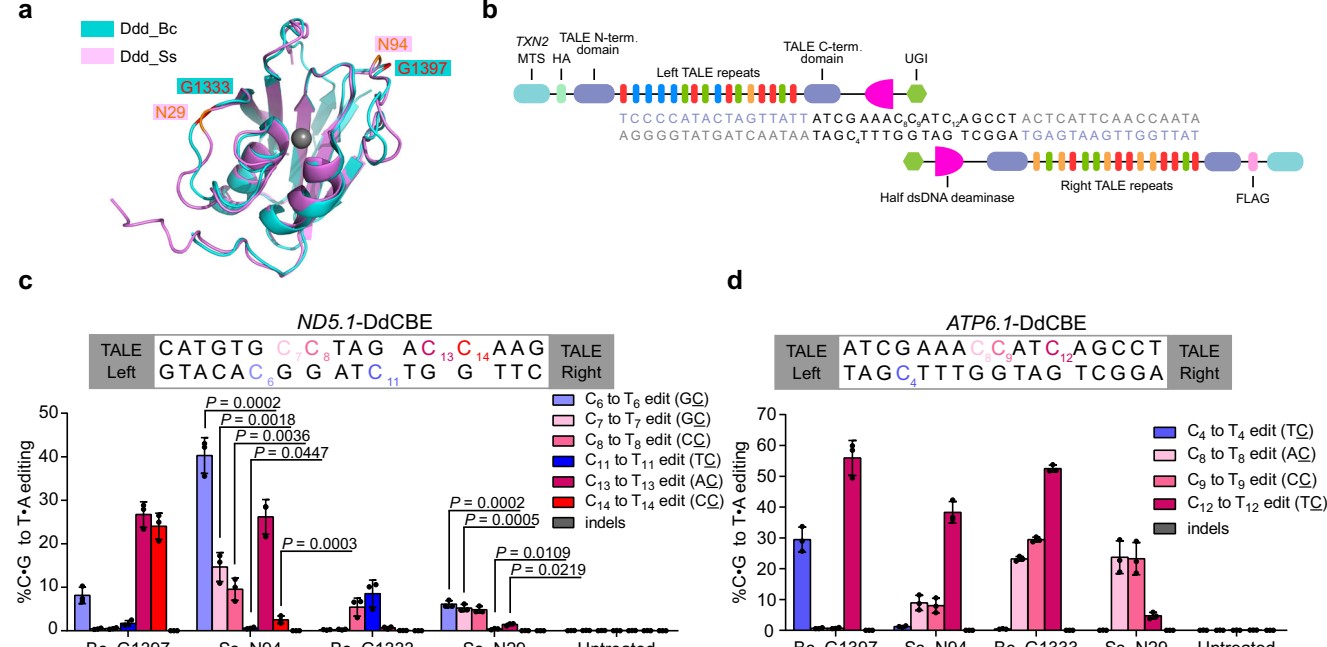

**Fig. 2 | DdCBE_Ss achieve high-efficiency editing at multiple mtDNA loci.**
**a** Structural alignment of Ddd_Bc (blue, PDB 6U08) and Ddd_Ss (purple, predicted by ColabFold). Ddd_Bc was split after the red-labeled amino acid G1333 or G1397, Ddd_Ss was split after the orange-labeled amino acid N29 or N94. **b** Split deaminases fused to left and right TALE proteins targeted to the *MT-ATP6* gene.
**c, d** mtDNA editing efficiencies and indel frequencies of HEK293T cells treated with

*ND5.1*-DdCBE (**c**) or *ATP6.1*-DdCBE (**d**). Bc_G1397: Ddd_Bc-G1397 split; Ss_N94: Ddd_Ss-N94 split; Bc_G1333: Ddd_Bc-G1333 split; Ss_N29: Ddd_Ss-N29 split. Shown are mean ± SD; *n* = 3 independent experiments. The transfection time was 3 days. For **c**, Ss_N94 and Ss_N29 was compared to Bc_G1397 and Bc_G1333, respectively. *P* values were calculated by Student's unpaired two-tailed *t*-test. Source data are provided as a Source Data file.

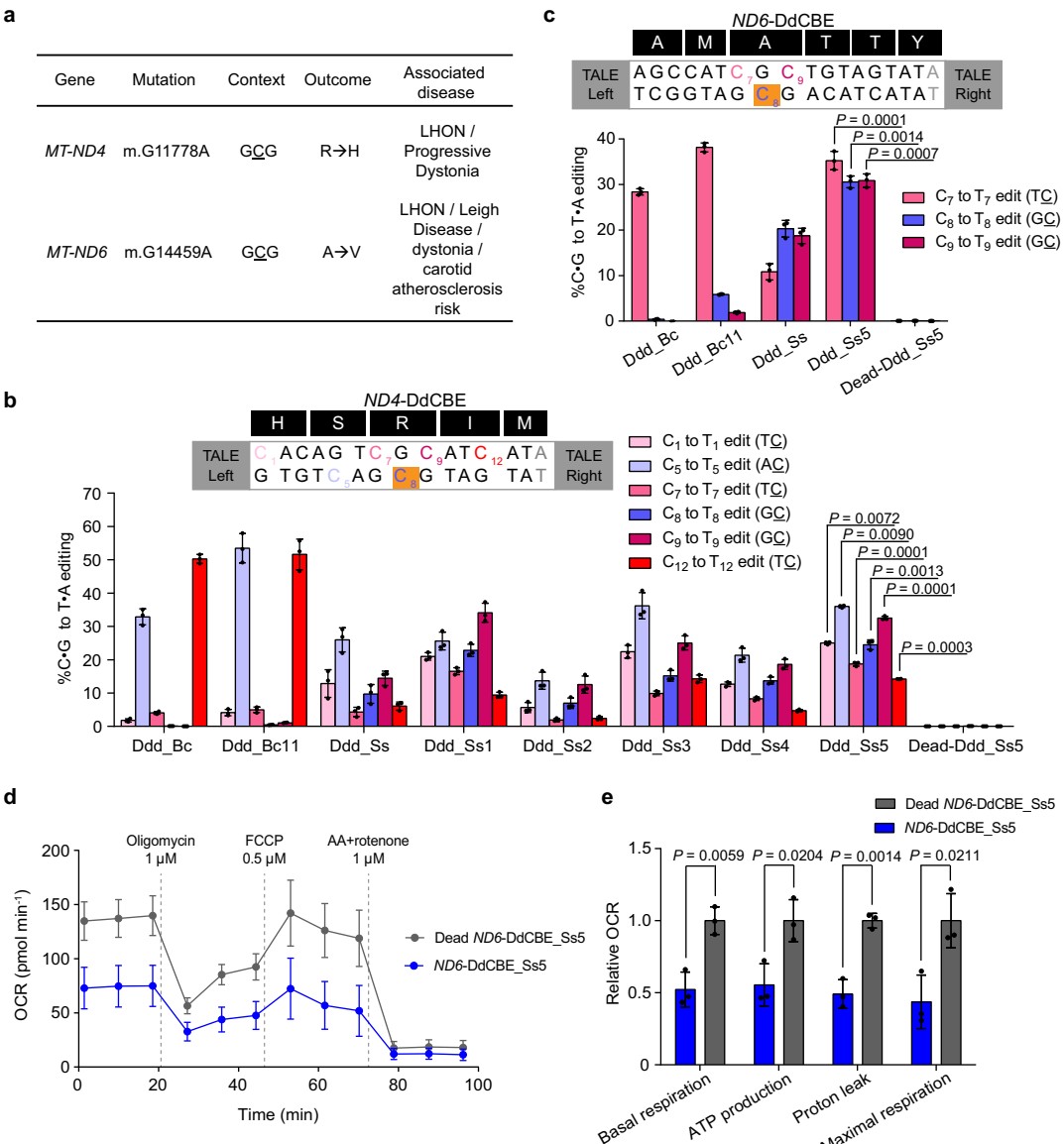

**Fig. 3 | DdCBE_Ss can install disease-associated mtDNA mutations in previously inaccessible GC sites. a** Use DdCBE_Ss to install disease-associated target mutations in human mtDNA (R, arginine; H, histidine; A, alanine; V, valine). **b, c** Mitochondrial base editing efficiencies of HEK293T cells treated with *ND4*-DdCBE (**b**), *ND6*-DdCBE (**c**), the background color of disease-associated target sites is orange, the gray nucleotide are part of the TALE binding site, Ddd_Ss1: Ddd_Ss (T26I + T77I + T110I); Ddd_Ss2: Ddd_Ss (T26I); Ddd_Ss3: Ddd_Ss (T77I); Ddd_Ss4: Ddd_Ss (T110I); Ddd_Ss5: Ddd_Ss (T77I + T110I); Dead-Ddd_Ss5:

Ddd_Ss (E44A + T77I + T110I). Shown are mean ± SD; *n* = 3 independent experiments. The transfection time was 3 days. DdCBE_Ss5 was compared against DdCBE_Ss. *P* values were calculated by Student's unpaired two-tailed *t*-test. **d, e** Oxygen consumption rate (OCR) (**d**) and relative values of respiratory parameters (**e**) in HEK293T cells treated with the *ND6*-DdCBE_Ss5 or dead *ND6*-DdCBE_Ss5. Shown are mean ± SD; *n* = 3 independent experiments. *P* values were calculated by Student's unpaired two-tailed *t*-test. Source data are provided as a Source Data file.

induced efficient editing at *ND5.1* and *ATP6.1* in HeLa and U2OS cells (Supplementary Fig. 6a–d). Cell viability assay showed that at least for the three sites studied, DdCBE_Ss did not show more significant toxicity than DdCBE_Bc (Supplementary Fig. 6e). Together, these results suggest that DdCBE_Ss is a general mtDNA editing tool with no apparent cell toxicity.

### Installment of disease-related mtDNA mutations with optimized DdCBE_Ss

Next, we tried to install two mutations[15,16] at GC context in *MT-ND4* and *MT-ND6* that have been linked to human diseases including Leber hereditary optic neuropathy (LHON) and Leigh syndrome (Fig. 3a), which are both devastating genetic diseases with no effective therapy. The DdCBE_Ss yielded around 10% editing for *MT-ND4* (C8) (Fig. 3b)

and 20% for *MT-ND6* (C8) (Fig. 3c). To further improve the performance of DdCBE_Ss, we adopted the mutations that have recently been shown to improve the editing efficiency of DdCBE_Bc[7] (T26I, T77I, and T110I corresponding to S1330I, T1380I, and T1413I). Excitingly, the DdCBE_Ss variant with T77I and T110I mutations (DdCBE_Ss5) significantly improved the editing efficiency at disease-associated target sites (~25% editing for *MT-ND4* C8 site and ~30% editing for *MT-ND6* C8 site) (Fig. 3b, c). However, like previously reported DdCBEs, DdCBE_Ss also caused many bystander mutations other than targeted sites (Supplementary Fig. 7). We then measured oxygen consumption rates for cells treated with *ND4*- or *ND6*-DdCBEs. Compared to control cells treated with catalytically inactive DdCBE, cells treated with *ND6*-DdCBE_Ss5 but not *ND4*-DdCBE_Ss5 showed lower rates of oxidative phosphorylation (Fig. 3d, e and Supplementary Fig. 8a, b). The lack of

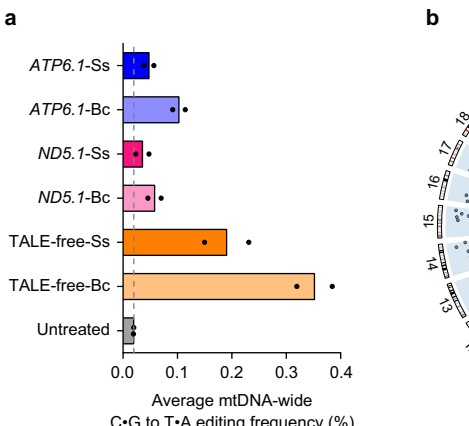

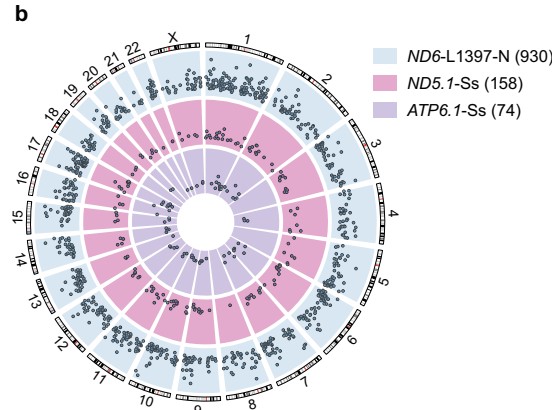

**Fig. 4 | Mitochondrial and nuclear genome-wide off-target editing by DdCBE_Ss. a** Average percentage of mtDNA-wide C•G to T•A off-target editing in untreated HEK293T cells and HEK293T cells treated with different DdCBEs. The vertical line represents the percentage of mtDNA C•G to T•A editing efficiency in untreated cells. Shown are the mean from $n = 2$ independent experiments.

**b** Genome-wide circos plots representing the distribution and Detect-seq scores of identified nuclear DNA off-target sites on each chromosome for three different DdCBEs. The number of off-target sites is shown in parentheses. Source data are provided as a Source Data file.

phenotype by *ND4*-DdCBE_Ss5 could be due to lower mtDNA editing frequencies in these cells (Supplementary Fig. 8c-f). These results suggest that our optimized DdCBE derived from Ddd_Ss can install disease-associated mtDNA mutations in previously inaccessible GC sites and mitochondrial mutations installed by DdCBE_Ss5 can cause biologically significant phenotypes.

## Characterization of mitochondrial and nuclear DNA off-target activities of DdCBE_Ss

We then characterized the off-target activities of Ddd_Ss. We performed an assay for transposase-accessible chromatin with sequencing (ATAC-seq) to detect off-target mutations in the mitochondrial genome. In all cases investigated, including *ND5.1*-DdCBE, *ATP6.1*-DdCBE, and TALE-free split Ddd pairs, DdCBE_Ss induced mutations at fewer off-target sites than DdCBE_Bc (Fig. 4a and Supplementary Fig. 9). In addition, we checked off-target editing by *ND4*-, *ND5.1*-, and *ND6*-DdCBE in pseudogenes encoded in the nuclear genome. For nuclear pseudogenes with the greatest homology (1–3 bp difference from mtDNA on-target sites), no significant off-target editing was observed (Supplementary Fig. 10). Recent work by ref. [17]. shows that the current design of DdCBEs could cause a broad extent of off-target editing at non-pseudogene sites in the nuclear genome. To comprehensively profile nuclear off-target editing activities of DdCBE_Ss, we performed Detect-seq experiment[18] for HEK293T cells transfected with plasmids encoding *ND5.1*-DdCBE_Ss, *ATP6.1*-DdCBE_Ss and *ND6*-L1397-N (DdCBE_Bc from Lei et al. as a positive control). Consistent with results from ref. [17], *ND6*-L1397-N caused editing at more than 900 off-target sites in the nuclear genome. *ND5.1*-DdCBE_Ss and *ATP6.1*-DdCBE_Ss caused editing at 158 and 74 off-target sites in the nuclear genome (Fig. 4b), at a range similar to other DdCBE_Bc constructs[17]. These data indicate that our mitochondrial DdCBE_Ss construct could induce numerous off-target editing in the nuclear genome. As demonstrated by previous studies[17,19], various approaches may be applied to reduce nuclear off-target editing by DdCBE, including fusion of nuclear export signals to DdCBE, co-expression of nucleus targeted inhibitor of DNA deaminases, or introduction of mutations to decrease the spontaneous assembly of split deaminase halves.

## Engineering DdCBE_Bc variant with improved activity and sequence compatibility

The DdCBE_Ss had a different sequence context preference from that of the DdCBE_Bc. Previous studies on C-to-U RNA editing enzyme APOBEC show that loop sequence is important for the sequence

preference of enzymatic activity[20]. To figure out which sequence variation may underlie the different substrate preferences of Ddd_Ss and Ddd_Bc, we mutated three loops proximal to the active site of Ddd_Bc individually according to loop sequences from Ddd_Ss (Supplementary Fig. 11a). For *MT-ND5.1* site, the editing efficiency was increased approximately 3.2-fold at C6 (GC context), 2.0-fold at C13 (AC context), and 2.3-fold at C14 (CC context) by mutant DdCBE_Bc with E1370N but not P1369T mutation in loop 2 (Fig. 5a and Supplementary Fig. 11b). In contrast, loop 3 modification had little impact on the activity of DdCBE_Bc, while loop 1 modification led to significantly reduced activity for DdCBE_Bc (Fig. 5a). Moreover, E1370N variant of DdCBE_Bc also yielded higher editing efficiency for *MT-ND1* and *MT-ND5.2* sites than the original DdCBE_Bc (Fig. 5b, c and Supplementary Fig. 11c, d). Together, these results show that the sequence compatibility and editing efficiency of DdCBEs may be rationally optimized by swapping sequences from different homologs.

## Discussion

In this study, we show that SPKK-related motifs at C-terminus are important for efficient DNA deamination by Ddd_Bc. We further identify numerous Ddd_Bc homolog candidates by PSI-BLAST and confirm four homologs with high dsDNA deaminase activity, all of which have SPKK-related motifs. We develop DdCBEs derived from Ddd_Ss and achieve high-efficiency editing for 14 mtDNA sites from 10 mitochondrial genes. Importantly, the engineered variant of DdCBE_Ss successfully installs disease-associated C-to-T mtDNA mutations with high efficiency at previously inaccessible GC context. Finally, by introducing variations from Ddd_Ss into Ddd_Bc, we successfully develop a DdCBE_Bc variant with broadened sequence compatibility and higher editing efficiency. Our study expands C-to-T mtDNA editing to previously inaccessible GC context and provides resources for further screening and engineering dsDNA base editors for a wide range of applications in biology and medicine.

## Methods

### Bacterial strains and culture conditions

All bacterial strains were grown in Luria-Bertani (LB) media or on Agar solidified LB media at 37 °C. Kanamycin (50 mg/L), Ampicillin (100 mg/L), L-arabinose (2 g/L), and IPTG (0.5 mM) were added to culture media when required. *E. coli* strains DH5α, BL21 (DE3), and BW25113 Δ*ung* were used for constructing and producing plasmids, protein expression, and heterologous expression of candidate deaminases to determine the substrate preference, respectively.

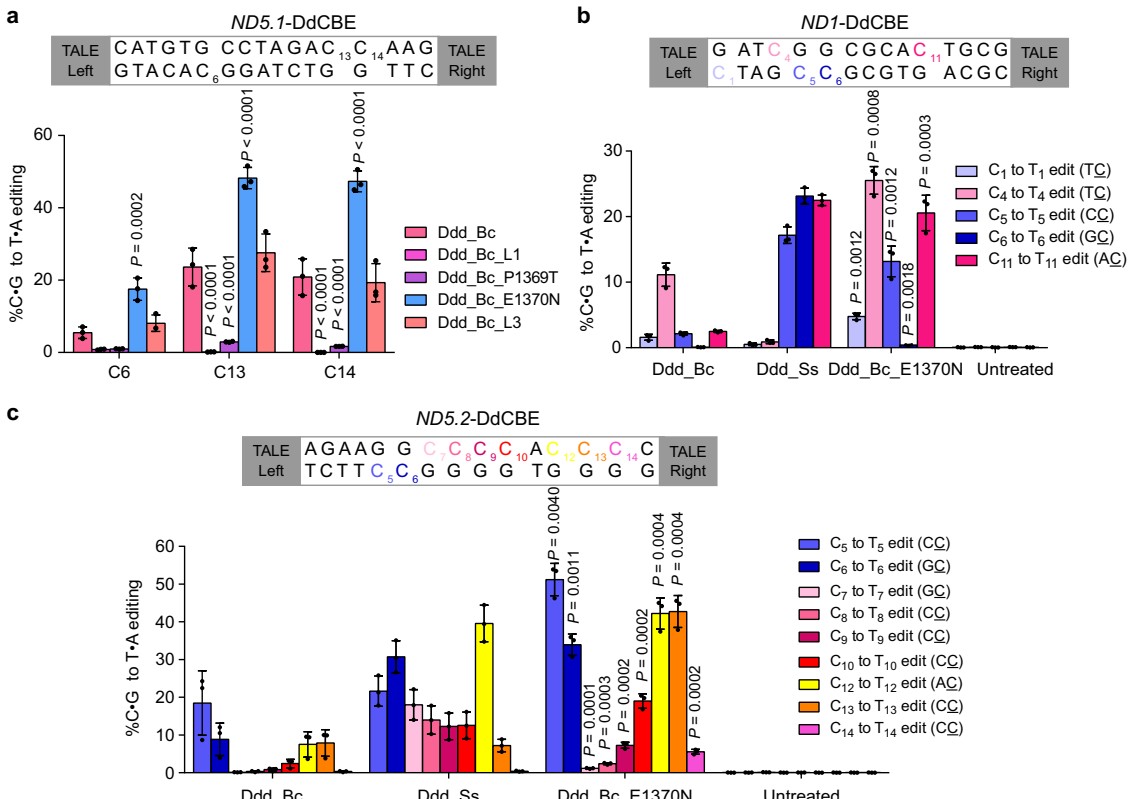

**Fig. 5 | E1370N mutation extensively increases the editing efficiency and sequence compatibility of DdCBE_Bc. a** mtDNA editing efficiencies of HEK293T cells treated with *ND5.1*-DdCBE derived from different Ddd_Bc variants. For Ddd_Bc_L1 and Ddd_Bc_L3, loop 1 and loop 3 sequences in Ddd_Bc are replaced with the corresponding sequence of Ddd_Ss as shown in Supplementary Fig. 11a. Shown are mean ± SD; *n* = 3 independent experiments. *P* values were calculated by two-way ANOVA followed by Dunnett's test. **b, c** mtDNA editing efficiencies of HEK293T cells treated with *ND1*-DdCBE (**b**) and *ND5.2*-DdCBE (**c**). Shown are mean ± SD; *n* = 3 independent experiments. Ddd_Bc_E1370N was compared against Ddd_Bc. *P* values were calculated by Student's unpaired two-tailed *t*-test. Source data are provided as a Source Data file.

## PSI-BLAST search and selection of candidate Ddd_Bc homologs

PSI-BLAST from the MPI Bioinformatics Toolkit[13] was used to identify homologs of Ddd_Bc. Briefly, the initial search of homologs was conducted by using the Ddd_Bc sequence as a query against the non-redundant (NR) protein database (2021, nr50_1_Nov). A query sequence profile was then constructed by including newly identified homologs for the next search iteration. The process was repeated until no new homologs are detected. The eight candidates were selected according to the PSI-BLAST score and C-terminal sequence diversity. For homologs with similar C-terminal sequences, only the one with the highest score was selected for further testing.

## Plasmid construction

To construct plasmids for deaminase expression, genes for deaminases and corresponding immunity proteins were synthesized by Azenta Life Sciences (Suzhou, China), cloned into the MCS-1 (BamHI and NotI sites, introducing an N-terminal hexahistidine tag) and MCS-2 (NdeI and XhoI sites) of pCOLADuet1. For deaminase expression[9] in *E.coli* BW25113 *Δung*, Ddd_Bc, Ddd_Ss, and Ddd_Fa were cloned downstream of araBAD promoter in pBAD, corresponding immunity protein genes driven by T7 promoter were cloned into the same plasmid downstream of the deaminase gene. For the assembly of DdCBEs, the golden gate cloning approach was used to assemble TALE arrays from tetramer templates[21], then TALE arrays were digested with NdeI (NEB) and BamHI (NEB) and ligated with split deaminases, UGI and other DdCBE sequences. DdCBE constructs in this study contained mitochondrial targeting sequences from the *TXN2* gene, Flag/HA tag, TALE array, split deaminases, and UGI. All DdCBE were fused to GFP through a self-cleaving T2A sequence except for Bc_G1397 and Ss_N94

in Fig. 2c, d. DdCBEs were cloned under the control of a CAGGS promoter in a piggyBac vector. Amino acid sequences to construct DdCBEs in this study are provided in Supplementary Notes 1, 2.

## Protein purification for DNA deamination assays in vitro

To express deaminases, BL21 (DE3) were transformed with pCOLA-Duet1 containing genes for a deaminase and its corresponding immunity protein. A single colony was picked for characterization and expansion. Protein expression was induced with 0.5 mM IPTG at $OD_{600}$ of 0.6–0.8 at 18 °C overnight. Cells were harvested and resuspended in lysis buffer (50 mM Tris-HCl, pH 7.5, 500 mM NaCl, 5% glycerol, 20 mM imidazole, 5 mM 2-Mercaptoethanol, and 1 mM PMSF) and lysed by sonication, the supernatant was separated by centrifugation in a JA-25.50 rotor (Beckman) for 30 min at 39,190 × *g*. The deaminase-immunity protein complex was purified from cell lysates by nickel affinity chromatography using 1 mL Ni Sepharose 6 Fast Flow agarose beads loaded onto a gravity-flow column (GE Healthcare). The supernatant was loaded onto the column and the resin was washed with 10 mL of wash buffer (50 mM Tris-HCl, pH 7.5, 500 mM NaCl, 20 mM imidazole, and 5 mM 2-Mercaptoethanol). The deaminase-immunity protein complex was eluted with 3 mL elution buffer (50 mM Tris-HCl pH 7.5, 300 mM imidazole, 500 mM NaCl, and 5 mM 2-Mercaptoethanol), then the deaminase is isolated from the complex through denaturing and renaturing steps. For denaturation, the eluted protein samples were added to 25 mL 6 M GuHCl denaturing buffer (50 mM Tris-HCl pH 7.5, 20 mM imidazole, 500 mM NaCl, and 5 mM 2-Mercaptoethanol) and incubated for 1 h at 4 °C. About 6 M GuHCl buffer with the eluted proteins were loaded on a gravity-flow column with 1 mL Ni Sepharose 6 Fast Flow agarose beads. The column was washed

with 10 mL 6 M GuHCl buffer to remove any remaining immunity protein. While still bound to Ni agarose beads, deaminase was renatured by sequential washes with 8 mL denaturing buffer with decreasing the concentration of GuHCl (5, 4, 3, 2, 1 M) containing 10 μM ZnCl$_2$, and a last wash with wash buffer to remove remaining traces of GuHCl. Proteins bound to the column were then eluted with 3 mL elution buffer. The eluted deaminases were purified again by size-exclusion chromatography using a Superdex75 column (GE Healthcare) in sizing buffer (20 mM Tris-HCl pH 7.5, 200 mM NaCl, 5 mM 2-Mercaptoethanol, and 5% glycerol). The fraction purity was evaluated by SDS−PAGE stained with Coomassie blue and the highest quality fractions were stored at −80 °C.

## DNA deamination assays
DNA deamination assays were conducted as reported previously[9], with some modifications. DNA substrates were purchased from Sangon Biotech (Shanghai, China), and contained a 6-FAM fluorophore at the 5′ end for visualization. To generate dsDNA substrates, a reverse complement oligo without modification was annealed to the substrate modified with a 6-FAM fluorophore at an equimolar concentration. Reactions were performed in 10 μL of deamination buffer consisting of 20 mM Tris-HCl pH 7.5, 200 mM NaCl, 5 mM 2-Mercaptoethanol, and 1 μM substrate. Reactions were incubated for 1 h at 37 °C, followed by the addition of 5 μL of UDG reaction solution (New England Biolabs, M0372S; 0.02 U/μL UDG in 1×UDG buffer) and an additional 30 min incubation at 37 °C. Cleavage of abasic sites generated by UDG-mediated cleavage of uracil residues in substrates was induced by the addition of 100 mM NaOH and incubation at 95 °C for 2 min. Reactions were analyzed by 20% acrylamide 8 M urea gel electrophoresis in 1×TBE buffer and the 6-FAM fluorophore signal was detected by fluorescence imaging with ChemiDoc MP imaging system (Bio-Rad). Quantification of the deamination percentage was performed via ImageJ. Uncropped gel images are provided in the Source Data file.

## Single nucleotide variation (SNV) analysis
To obtain genome DNA, *E.coli* BW25113 Δ*ung* strains expressing candidate deaminases were inoculated into 20 mL of LB broth in a 1:100 dilution and cultures were grown to approximately OD$_{600}$ 0.6, then deaminases was induced with 2 g/L L-arabinose for 1 h. A 3 mL bacteria culture was used to extract the bacteria genome with FastPure Blood/Cell/Tissue/Bacteria DNA Isolation Mini Kit (Vazyme, DC112). The extraction yield was quantified using Qubit®3.0 Fluorometer (Thermo Fisher Scientific). Sequencing libraries were constructed using the VAHTS Universal Plus DNA Library Prep Kit for Illumina V2 (Vazyme, ND627) following the manufacturer's instructions, except that the VAHTS HiFi Amplification Mix was substituted with KAPA HiFi HotStart Uracil+ ReadyMix (Kapa Biosystems, KK2801) to enable efficient amplification of uracil encountered in the DNA template. Library concentration and quality were evaluated with Qubit and 1% agarose gel electrophoresis. Sequencing was performed with an Illumina Nova-seq 6000 sequencing system (Novogene), and the BWA software (version 0.7.17) was used to map reads against a reference genome (NC_000913.3, https://www.ncbi.nlm.nih.gov/nuccore/NC_000913.3). Duplicates were removed using Picard tools (version 2.18.29). Pileup data from alignments were generated with SAMtools (version 1.14), and variant calling was performed with VarScan (version 2.4.4)[22]. A threshold for SNV validation of variant frequency >0.01, coverage >50 reads per base, and *p* value <0.01 was used. Probability logos of the consensus region flanking modified bases were generated with the online tool WebLogo 3 (version 3.7.12, https://weblogo.threeplusone.com).

## Mammalian cell culture and transfection
HEK293T (ATCC, CRL-3216), HeLa (ATCC, CCL-2), and U2OS (ATCC, HTB-96) cells were cultured at 37 °C under 5% CO$_2$ in high glucose Dulbecco's Modified Eagle's Medium (DMEM, Hyclone, D6429) supplemented with 10% (v/v) FBS (PANSera, 2602-P130707) and 1% (v/v) penicillin/streptomycin (Gibco, 15140163). Cells were passaged every 2–3 days. For HEK293T transfection, $8 \times 10^4$ cells were plated on 0.01% (w/v) poly-D-lysine (Beyotime, ST508) coated 24-well plate around 20 h before transfection. For HeLa and U2OS transfection, $6 \times 10^4$ cells and $4 \times 10^4$ cells were respectively plated on a 24-well plate around 20 h before transfection. Cells were transfected with 250 ng of each DdCBE monomer to make up 500 ng of total plasmid DNA using jetPRIME transfection reagent (Polyplus-transfection, 468 PT-114-75). Cells were harvested 72 h after transfection for genomic DNA extraction, then the region containing the targeted editing site was PCR amplified for Sanger sequencing (Azenta Life Sciences) or for producing libraries for next-generation sequencing.

## Targeted amplicon sequencing and analysis
For targeted amplicon sequencing, the region of interest was first amplified by PCR round 1 using KAPA HiFi HotStart Uracil+ ReadyMix (Kapa Biosystems, KK2801), then PCR round 1 product were reamplified in PCR round 2 to add Illumina indices using N323 VAHTS RNA Multiplex Oligos Set 1 for Illumina (Vazyme, N323). The number of PCR cycles was optimized using qPCR to the top of the linear range to minimize biased amplification. Typically, we used 12 cycles in PCR round 1 and 10 cycles in PCR round 2 for 100 ng whole-cell genome as initial templates. The products were further purified by 1×VAHTS DNA Clean Beads (Vazyme, N411), then the libraries were subjected to sequence by an Illumina Nova-seq 6000 sequencing system. CRISPResso2 (version 2.2.7) was used for targeted amplicon sequencing analysis[23]. The output file "Nucleotide_percentage_table.txt" was imported into Microsoft Excel 2019 for the quantification of editing frequencies. Adobe Illustrator CC2018 was used for data visualization. Each amplicon was sequenced with more than 10,000× coverage. Primers for PCR of on-target and off-target regions are listed in Supplementary Tables 1, 2.

## Cell viability assay
Cell viability was measured every 3 to 6 days over an 18-day time course using the CellTiter-Glo 2.0 Assay (Promega, G9241) following the manufacturer's instruction. Luminescence was measured in 96-well flat-bottom white polystyrene microplates (Corning, 3917) using Centro XS$^3$ LB 960 reader with a 1 s integration time.

## Oxygen consumption measurements
HEK293T cells were transfected with the left and right halves of DdCBE fused to GFP and mCherry through a self-cleaving T2A sequence. Three days after transfection, cells that express both halves of the DdCBE were enriched by fluorescence-activated cell sorting (FACS) using BD FACSAria III. The FACS gating strategy is shown in Supplementary Note 3. The sorted cells were recovered three generations (passaged every other day) before measuring the oxygen consumption rate. Oxygen consumption rates of HEK293T cells were measured using a Seahorse XF24 Extracellular Flux analyzer (Agilent) following the manufacturer's instruction. In brief, 45,000 cells were seeded on the Seahorse plate coated with 0.01% (w/v) poly-D-lysine for 16 h. Analysis was performed in Seahorse XF DMEM Medium pH 7.4 (Agilent, 103575-100) supplemented with 10 mM glucose (Agilent, 103577-100), 2 mM L-glutamine (Agilent, 103579-100), and 1 mM sodium pyruvate (Sigma, S8636). Mito stress protocol was applied with the use of 1 μM oligomycin, 0.5 μM FCCP, and 1 μM rotenone + antimycin A.

## ATAC-seq
ATAC-Seq was performed as previously described in ref. [24] with slight modifications. Cells were digested and washed with cold PBS, and 50,000 cells were collected by centrifugation at $500 \times g$ for 5 min at 4 °C. Cell pellets were lysed in 50 μL of cold lysis buffer (10 mM Tris-

HCl, pH 7.4, 10 mM NaCl, 3 mM $MgCl_2$, 0.1% IGEPAL CA-630) and incubate on ice for 5 min. Centrifuge immediately after lysis at $500 \times g$ for 10 min at 4 °C and the cell pellets were tagmented with 50 μL transposition mix (10 μL 5 × TTBL, 5 μL TTE Mix V50, 35 μL nuclease-free $H_2O$, Vazyme TD501). Samples were incubated at 37 °C for 30 min on a PCR thermocycler then purified with VAHTS DNA Clean beads (Vazyme, N411) and eluted in 24 μL $ddH_2O$. All 24 μL of the eluate was amplified using 10 μL 5 × TAB, 5 μL PPM, 5 μL N5 Primer, 5 μL N7 Primer, 1 μL TAE (Vazyme, TD501) in a total volume of 50 μL using the following protocol: 72 °C for 5 min, 98 °C for 30 s and then 12 cycles of (98 °C for 15 s, 60 °C for 30 s, and 72 °C for 3 min), followed by a final 72 °C extension for 5 min. Amplification products were further purified and size selected by VAHTS DNA Clean Beads. The libraries were then subjected to sequence by an Illumina Nova-seq 6000 sequencing system.

### Analysis of mitochondrial genome-wide off-target editing

Analysis was conducted as reported previously[7,8], with some modifications. Genome mapping was performed with BWA software (version 0.7.17) using NC_012920.1 genome as a reference (https://www.ncbi.nlm.nih.gov/nuccore/NC_012920.1). Duplicates were marked using Picard tools (version 2.18.29). The sequencing depth of each sample was more than 3000×. To calculate the mtDNA-wide average C•G to T•A editing frequency, REDItools (version 1.2.1) was used[25]. All bases except cytosines and guanines were removed, and bases with a PHRED quality score greater than 30 was calculated. The on-target sites (the region between two TALE binding sites) were excluded to consider only off-target effects. C•G to T•A SNVs present at high frequencies (>50%) in untreated samples (regarding as SNVs in the cell lines) were also excluded. The remaining bases with C•G to T•A editing frequency greater than 0.1% was counted as off-target sites. The average off-target editing frequency was then calculated as the sum over all off-target sites C•G to T•A editing frequency divided by the total number of all non-target C•G bases. mtDNA-wide graphs were created by plotting the editing efficiency at on-target and off-target sites with an editing frequency greater than 1% across the entire mitochondrial genome.

### Detect-seq experiment and analysis

Detect-seq experiment was conducted as reported previously[18]. About $4 \times 10^5$ HEK293T cells were plated on a six-well plate and 1250 ng of each DdCBE monomer was used for transfection. Cells were harvested 72 h after transfection, and genomic DNA was isolated using DNA Isolation Mini Kit (Vazyme, DC112). DNA damage was repaired or protected to reduce the background noise. DNA fragments containing DdCBE-induced deoxyuridine bases were recognized by UDG and labeled via nick translation and subsequent chemical reactions. The biotin-labeled fragments were enriched by streptavidin C1 beads (Invitrogen, 65001) and ligated with Y adapters. Detect-seq libraries were sequenced on MGISEQ-2000. The efficiency and specificity of Detect-seq libraries were evaluated by qPCR and Sanger sequencing on spike-in molecules. Detect-seq data were mapped and processed following the steps described in the Detect-seq tool website (www.detect-seq.com). The DdCBE default parameters were used for analysis.

### Statistics and reproducibility

Graphical visualization of data and statistical analyses were performed with GraphPad Prism 6. The data were presented as mean ± SD except where indicated otherwise. Student's unpaired two-tailed $t$-test and two-way ANOVA with Dunnett's test were used as specified in the figure legends. $P$ value <0.05 was considered statistically significant.

### Reporting summary

Further information on research design is available in the Nature Portfolio Reporting Summary linked to this article.

## Data availability

The high-throughput sequencing data generated in this study have been deposited in the NCBI's Sequence Read Archive (SRA) database under accession code PRJNA915236. NC_000913.3 was used as the *E.coli* reference genome. NC_012920.1 was used as a human mitochondrial reference genome. Amino acid sequences to construct DdCBEs in this study are provided in Supplementary Notes 1 and 2. Source data are provided with this paper.

## Code availability

Bacterial genome SNVs were called using VarScan2 (version 2.4.4)[22]. High-throughput sequencing data of targeted amplicon was analyzed using CRISPResso2 (version 2.2.7)[23]. mtDNA off-target effects from ATAC-seq data were calculated using REDItools (version 1.2.1)[25]. Detect-seq data were processed using the Detect-seq tool (www.detect-seq.com).

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

## Acknowledgements
We would like to thank members of Wang's laboratory and Dr. Chengqi Yi for critical reading and discussion of the project. We thank Chengying Ma and Hongwei Li for their help with protein purification and biochemical assays. We thank the National Center for Protein Sciences at Peking University for assistance with imaging, FACS, and sequencing; Huan Yang for assistance with FACS; Guilan Li and Xiaohui Zhang for assistance with NGS experiments. This study was supported by The National Key Research and Development Program of China [2021YFA0100200 and 2018YFA0107601] and the National Natural Science Foundation of China [91940302, 32130017, and 32025007] to Y.W.

## Author contributions
Y.W. supervised the research and wrote the manuscript with L.M.'s and M.S.'s assistance. Y.W. and L.M. designed the study. L.M., M.S., and Y.-X.L. performed most of the experiments. L.M. and G.X. performed bioinformatics analysis and made all the figures. X.R. constructed Detect-seq libraries. D.W., A.C., and N.G. provided reagents and assistance with protein purification experiments. M.N., F.X., and X.W. facilitated mitochondrial function analysis. Y.Y. and W.W. provided reagents and facilitated TALE construction.

## Competing interests
Y.W. and L.M. have filed a patent application on DdCBE_Ss and its derivatives. The remaining authors declare no competing interests.
