## [Peer Review File · Nature Communications]

Reviewers' Comments:

Reviewer #1:

Remarks to the Author:

Using classic bioinformatic searching, the authors identify gene homologous to the DddA-tox base editor used in mitochondrial editing since the Mok et al, 2020 paper. With this technique the identify a homologue from *Simiaoa sunii*. Testing of these proteins reveals a DC editing context, finally allowing for the GC editing that is missing in the current editors available.

The expression experiments, and proof-of-concept demonstration of mutations are carried out in ways similar to the earlier reports of these technologies, so appear sufficient. The study into the SPKK motif is novel, and a helpful mechanistic insight that may be expanded on or used in the future.

The report is clear, brief and to the point. Numerical comparisons to other editors on only qualitatively described, so the authors may wish to consider if difference should be more formally tested through statistics.

My comments;

First – are the sites selected for mutagenesis represented by NUMTs? Could this be an unaccounted for variable that needs consideration? As with all mtDNA sequencing papers, the NUMTs need to be at least discussed so that we readers understand the context of the data provided.

Secondly, some recent work on these types of base editors have revealed some off-target, nuclear activity (<https://www.nature.com/articles/s41421-022-00391-5> & <https://www.nature.com/articles/s41586-022-04836-5>). Given the profile of these comments, I felt that the authors should at least mention this as a potential pitfall. While I feel that these issues are most likely a construct design issue, and not an inherent hazard of the technology, the authors could simply suggest construct designs to limit these issues (ex. including nuclear export signals for therapeutic use, etc).

Finally, there is no discussion of true off-target mutation assays, from what I could observe in the manuscript. Due to the nature of the sequencing method used, I am not sure if this was adequately addressed. If the authors have sequence information of other C's outside of the target window between the TALEs, this information would be greatly appreciated in the supplementals. Otherwise, it may be good to check for this. Of course, off-target issues are more likely to be an issue of TALE design, but it would good to know if this enzyme is as specific as the classic DddA.

Minor editing issues I noticed.

Check the grammar of the sentence in lines 48 – 50, or split it into a few sentences for more clarity

Line 61 – “*Simiaoa Sunii*” should be written “*Simiaoa sunii*”.

Reviewer #2:

Remarks to the Author:

This manuscript by Mi, Shi, Wang and colleagues explores natural homologues of the dsDNA deaminase DddA with expanded sequence targeting. Mitochondrial base editing has been recently advanced by the coupling of the bacterial toxin DddA from *B. cenocepacia* (Ddd-Bc) with TALE domain, which can help target specific genomic sites, especially in mitochondria. The existing toolbox could initially edit at TC sites, with inclusion of AC and CC sites with more recent directed evolution. In this manuscript, the authors aimed to explore activity and sequence targeting in the larger DddA family with the hopes of expanding to include GC sites as well.

The work is grounded in some initial biochemical exploration, whereby a C-terminal SPKK motif was suggested to support dsDNA deaminase activity. From >500 identifiable homologs of DddA, the authors focus on a group of 8 and note that in vitro activity largely tracked with the presence of this C-terminal motif in 4. Overexpression in *E. coli*, revealed a broader sequence context

preference for some, including deamination at GC contexts with the DddA homologue from *S. sunii* (Ddd-Ss). Given the evidence for broader sequence preferences, the authors picked split sites based on prior work with Ddd-Bc and fused halves of the deaminase each to a TALE. The authors initially edit a ND5 mitochondrial site and compare Ddd-Bc and DDD-Ss with different split sites, where both constructs can edit, but the specific C bases changed in the editing window differ between constructs, findings replicated at a second mitochondrial site (ATP6). Across 8 broader sites average editing by the Ddd-Bc construct was ~22% while it is 28% with Ddd-Ss (Supp Table 2), with some sites accessed preferentially by one construct over the other. Focusing on several sites where GC edit might be relevant to disease, the authors compare constructs, showing that both constructs can edit in these windows, but that there is a higher rate of editing at the GC site, and that editing can be improved by adding mutations to Ddd-Ss that have been shown to improve Ddd-Bc.

Overall, this work adds a new dsDNA deaminase family member for incorporation into the genome editing toolbox. Some of the mechanistic and biochemical conclusions, such as the attribution of function to the C-terminal SPKK motif and the quantitative rigor of sequence specificity profiling, require more substantiation. More generally, the applications to genome editing suggest that the Ddd-Ss behaves differently than the Ddd-Bc construct given its different sequence preference, however it would be an overreach to say that this advance fills a substantial gap in the field or that this tool will potentiate many experiments that could not have been done otherwise. Below are a few key notes regarding precedent and impact, along with experimental and other points that would be important to address. These include:

1) Impact of broader exploration of dsDNA deaminase family. A major aspect of the claim towards innovation in the manuscript is the exploration of the larger dsDNA family, with preliminary characterization of eight family members and advancement of one into genome editing. In this realm there are two precedents worth noting. As appropriately cited, recent work (Mok, 2022) has expanded the Ddd-Bc specificity beyond the initial TC to include AC and CC, thus leaving only GC as an opportunity for further expansion in this work. Second, Mougous and colleagues have looked into the broader dsDNA deaminase family (de Moraes et al, eLife, 2021) in work which was not cited here. Alternative sequence preferences were one of the findings with other family members, including an analog with activity on ssDNA, but also dsDNA activity, albeit less substantial. In the bigger picture, recent and longer standing studies on various DNA deaminases have established that each has its own sequence preference and that these can readily be altered, making the report here interesting but also somewhat incremental.

2) Impact of broadening to include GC targeting. The Ddd-Ss does appear to have added the ability to target GC relative to the Ddd-Bc. A challenge with these dsDNA genome editors is that they can make multiple mutations in the same editing window, thus there is likely a very narrow set of targets for which editing of a single GC site can be achieved and is of high scientific or clinical value. In their example loci, the authors offer evidence for the fact that they have increased editing at a 'previously inaccessible' site, it was less clear that they 'reversed' a mutation specifically without local bystander edits. Perhaps by showing the CRISPResso outputs (analogous to Supp Fig S3) for the loci in Fig 2e-f it will be clear if they can edit the ND4 and ND6 loci without bystander edits to make the desired mutational change only (without bystander edits). Similarly, whether mutations introduced could be biologically valuable was not established, as there was no characterization of the editing outcomes on mitochondrial function, only sequencing based assessment.

3) Off target activities. The dsDNA deaminase editors have raised concerns with the extent of off target activities that can occur (Lei et al, Nature 2022). Despite deaminase enzyme splitting, a significant amount of C to T transitions were observed in genomic DNA. The introduction of a new dsDNA tool comes with some obligations to similarly profile the extent of off-target effects, particularly when the construct may be more active and target different sequences. Such off-target analysis was absent in the current manuscript and would impact whether others would choose to employ these new constructs in their future work.

Other points to address:

1) The authors postulate that there is a critical role for the C-terminal SPKK motif in dsDNA

recognition, attributing DNA binding activity to this region. However, an examination of the structure (PDB 6U08) reveals that the C-terminal region is on the opposite face of the enzyme from the DNA binding catalytic face. It is equally likely that the C-terminal region has some structural role or other role, and nothing to do with 'minor groove DNA binding'. The authors should provide direct evidence for this postulated role if this conjecture is to be made.

2) Fig 1c shows quantified data and states WT data is from n = 2 replicates. If this is the case, error bars should likely not be shown. Also, in the legend with corresponding gels in Supp Fig S1 it states that all gels were done with n = 3 rather than n = 2. Please clarify and adjust.

3) The experimental setup in Supp Fig 2 has some issues with regard to demonstrating sequence specificity. The substrate is of a sequence 5'-FAM-N6-TCACGCC-N8'-3'. On the gel, the only thing that will appear after deamination and UDG treatment is the 5'-fragment given the 5'-FAM. Thus, if Ddd-Bc is much better at TC sites as expected, it will mask any activity at the other sequence contexts. Although the overall data, particularly with E. coli bases assays help to support the overall conclusion, this specific assay as presenting could be misleading and should be repeated with a substrate that changes the order of the TC, AC, GC and CC sites from 5'- to 3' direction, or four substrates each of which has a different NC sequence. This will allow for more direct quantification and comparison of the enzymes.

4) Fig 1d deaminase activity column and Supp Fig 2. The data should be presented in a quantified manner. The data in Fig 1d are provided as weak, * or ** for lower, comparable versus higher activity. Implied is that this is based on a single data point from either 0.5 µM or 10 µM enzyme, although this is not clear. Quantitative analysis should be provided for each construct analogous to Fig 1c, given that the assay was performed in a quantitative format.

5) Fig 2 and SI Table 2. In the SI Table, is the % Editing reported based on total editing across C sites in the locus of interest or based on a specific C site? As noted above, it is important to highlight whether edits are happening at specific sites or if bystander edits are common. The CRISPResso output should be provided for the sites analyzed by NGS and ideally all of the amplicons should be studied by NGS and not by Sanger estimation of editing (as Sanger does not allow for analysis of whether there are multiple edits in single loci).

6) A very minor point, but given Genus species (Capital, lowercase) nomenclature and typical descriptions of such constructs in the field, it would seem more appropriate to refer to Ddd-Ss rather than Ddd-SS. On a related note, it may be easier to follow the experiments and comparisons if the prior Ddd is referred to as Ddd-Bc rather than DddAtox as they are both DddAtox, just from different species.

Reviewer #3:

Remarks to the Author:

Mi et al. present DdCBEs containing DddA_SS, a DddA homolog from Simihoa Sunii, enabling mitochondrial DNA editing in a previously inaccessible GC context. The authors found that the SPKK motif at the C terminus of DddA homologs are essential for deaminase activity in vitro and that DddA_SS catalyzes cytosine deamination at GC context. They used DdCBEs containing DddA_SS to install mitochondrial disease-associated mutations in human cells. This study expands mitochondrial DNA editing and provides novel therapeutic opportunities. I have the following points to improve this manuscript.

1. What is the basis of choosing eight candidates among 555 DddA homologs?
2. The authors showed that the E1370N mutation in a loop region increases editing efficiency and sequence compatibility but did not explain the rationale behind this particular mutation. Was it chosen among (how) many mutants in the three loop regions?
3. The authors need to profile off-target activity of DdCBE_SS in mtDNA at minimum and also in genomic DNA. It is also important to check whether DdCBE_SS is cytotoxic in comparison with

DdCBEs.

4. The authors need to provide full DNA sequences of DdCBE_SS including TALE sequences.
5. How many cell lines were used to show the activity of DdCBE_SS. At least two different cell lines must be used.
6. Most of Supplementary Figures must be shown as main figures. There are only two main figures.

Response to Reviewer Comments:

We sincerely thank the three reviewers for their careful analysis and constructive comments and suggestions for improving the manuscript. To address the issues raised by the reviewers, we have now included new experimental data. Please find below our point-by-point responses to reviewers' comments (texts in light blue). In addition, text changes in manuscript are highlighted in yellow shade.

Reviewer #1 (Remarks to the Author):

Using classic bioinformatic searching, the authors identify gene homologous to the DddA-tox base editor used in mitochondrial editing since the Mok et al, 2020 paper. With this technique the identify a homologue from *Simiaoa sunii*. Testing of these proteins reveals a DC editing context, finally allowing for the GC editing that is missing in the current editors available.

The expression experiments, and proof-of-concept demonstration of mutations are carried out in ways similar to the earlier reports of these technologies, so appear sufficient. The study into the SPKK motif is novel, and a helpful mechanistic insight that may be expanded on or used in the future.

The report is clear, brief and to the point. Numerical comparisons to other editors on only qualitatively described, so the authors may wish to consider if difference should be more formally tested through statistics.

Answer: We sincerely thank the reviewer's comments and valuable suggestions for enhancing the quality of our manuscript. We have now added statistical test when a comparison is made, including Figures 2, 3 and 5.

My comments;

First – are the sites selected for mutagenesis represented by NUMTs? Could this be an un-accounted for variable that needs consideration? As with all mtDNA sequencing papers, the NUMTs need to be at least discussed so that we readers understand the context of the data provided.

Answer: We included discussion and new data for this part. We used the off-target prediction tool PROGNOS (Fine, E. J., et al. Nucleic Acids Research 2014) to predict nuclear off-target sites of *ND4*-, *ND5.1*-, and *ND6*-DdCBE, and find pseudogenes *MTND4P12*, *MTND5P11* and *MTND6P4* have the greatest homology, only 1-3 bp different from the on-target sites. We then used targeted amplicon sequencing to characterize potential off-target editing at these NUMTs and did not detect any off-target editing events for all three sites (**Figure R1**). These data are presented in Supplementary Figure 10.

Figure R1. Undetectable off-target editing activity on nuclear pseudogenes by *ND4*-, *ND5.1*-, and *ND6*-DdCBEs.

a-c, The on-target editing site in mtDNA and the corresponding sites in nuclear DNA with the greatest homology are shown for *ND4*-DdCBE (**a**), *ND5.1*-DdCBE (**b**), and *ND6*-DdCBE (**c**). TALE binding sites are shown in purple. Target cytosines are in blue. Nucleotide mismatches between the mtDNA and nuclear pseudogene are in red. Shown are mean \pm SD; $n = 3$ independent experiments.

Secondly, some recent work on these types of base editors have revealed some off-target, nuclear activity (<https://www.nature.com/articles/s41421-022-00391-5> & <https://www.nature.com/articles/s41586-022-04836-5>). Given the profile of these comments, I felt that the authors should at least mention this as a potential pitfall. While I feel that these issues are most likely a construct design issue, and not an inherent hazard of the technology, the authors could simply suggest construct designs to limit these issues (ex. including nuclear export signals for therapeutic use, etc).

Answer: Thank the reviewer for this comment. We are lucky to have the help from Yi Lab who invented Detect-seq and published one of the above-mentioned studies (Lei et al. Nature, 2022). We now included the Detect-seq data (**Figure R2a**, also shown as Fig. 4b in the revised manuscript). As shown below, DdCBE_Ss induced numerous mutations in nuclear genome, but not more than numbers reported for DdCBE_Bc (**Figure R2b**). We have added the data and discussion in the maintext as following:

To comprehensively profile nuclear off-target editing activities of DdCBE_Ss, we performed Detect-seq experiment for HEK293T cells transfected with plasmids encoding *ND5.1*-DdCBE_Ss, *ATP6.1*-DdCBE_Ss and *ND6*-L1397-N (DdCBE_Bc from Lei et al. as a positive control). Consistent with results from Lei et al., *ND6*-L1397-N caused editing at more than 900 off-target sites in nuclear genome. *ND5.1*-DdCBE_Ss and *ATP6.1*-DdCBE_Ss caused editing at 158 and 74 off-target sites in nuclear genome (**Fig. 4b**), at a range similar to other DdCBE_Bc constructs. These data indicate that our mitochondrial DdCBE_Ss construct could induce numerous off-target editing in nuclear genome. As demonstrated by previous studies, various approaches may be applied to reduce nuclear off-target editing by DdCBE, including fusion of nuclear export signals to DdCBE, co-expression of nucleus targeted inhibitor of DNA deaminases, or introduction of mutations to decrease the spontaneous assembly of split deaminase halves.

Figure R2. Nuclear off-target editing activities of DdCBE_Ss measured by Detect-seq.

(a) Genome-wide circos plots representing the distribution and Detect-seq scores of identified nuclear DNA off-target sites on each chromosome for three different DdCBEs. The number of off-target sites is shown in parentheses. (b) Results copied from Lei et al. for comparison.

Finally, there is no discussion of true off-target mutation assays, from what I could observe in the manuscript. Due to the nature of the sequencing method used, I am not sure if this was adequately addressed. If the authors have sequence information of other C's outside of the target window between the TALEs, this information would be greatly appreciated in the supplementals. Otherwise, it may be good to check for this. Of course, off-target issues are more likely to be an issue of TALE design, but it would good to know if this enzyme is as specific as the classic DddA.

Answer: In the revision, we performed assay for transposase-accessible chromatin with sequencing (ATAC-seq) to detect off-target mutations in mitochondrial genome. As shown in **Figures R3** and **R4**, DdCBE_Ss appear to be a little more specific than DdCBE_Bc in all cases investigated, including *ND5.1*-DdCBE, *ATP6.1*-DdCBE and TALE-free split Ddd pairs. These data were shown in Figure 4a and Supplementary Figure 9.

Figure R3. Mitochondrial off-target activities measured by ATAC-seq. Average percentage of mtDNA-wide C•G to T•A off-target editing in untreated HEK293T cells and HEK293T cells treated with different DdCBEs. The vertical line represents the percentage of mtDNA C•G to T•A editing frequency in untreated cells. Shown are means from n = 2 independent experiments.

Figure R4. The C•G to T•A editing frequency in mitochondrial genome of HEK293T cells treated with DdCBE_Bc or DdCBE_Ss.

a-f, Average C•G to T•A editing frequency of on-target (red dots) and off-target (gray dots) sites across mtDNA are shown for HEK293T cells treated with (a) DdCBE_Bc without TALE arrays, (b) DdCBE_Ss without TALE arrays, (c) *ND5.1*-DdCBE_Bc, (d) *ND5.1*-DdCBE_Ss, (e) *ATP6.1*-DdCBE_Bc, and (f) *ATP6.1*-DdCBE_Ss. Sites with average editing frequency greater than 1% are shown. Data are shown as means from $n = 2$ independent experiments.

Minor editing issues I noticed.

Check the grammar of the sentence in lines 48 – 50, or split it into a few sentences for more clarity

Answer: We have changed the sentence in lines 48-50 to: “Next, we used PSI-BLAST to identify homologs of Ddd_Bc. We successfully identified 555 candidate homologs of Ddd_Bc from the non-redundant protein database (nr50_1_Nov, 2021)”. In addition, we added a new section in methods to describe details of PSI-BLAST search and criteria used to select 8 candidates for activity testing.

Line 61 – “Simiaoa Sunii” should be written “Simiaoa sunii”.

Answer: Changes are made as suggested throughout the text and figures. Accordingly, Ddd_SS and DdCBE_SS are changed to Ddd_Ss and DdCBE_Ss. Thanks!

Reviewer #2 (Remarks to the Author):

This manuscript by Mi, Shi, Wang and colleagues explores natural homologues of the dsDNA deaminase DddA with expanded sequence targeting. Mitochondrial base editing has been recently advanced by the coupling of the bacterial toxin DddA from *B. cenocepacia* (Ddd-Bc) with TALE domain, which can help target specific genomic sites, especially in mitochondria. The existing toolbox could initially edit at TC sites, with inclusion of AC and CC sites with more recent directed evolution. In this manuscript, the authors aimed to explore activity and sequence targeting in the larger DddA family with the hopes of expanding to include GC sites as well.

The work is grounded in some initial biochemical exploration, whereby a C-terminal SPKK motif was suggested to support dsDNA deaminase activity. From >500 identifiable homologs of DddA, the authors focus on a group of 8 and note that *in vitro* activity largely tracked with the presence of this C-terminal motif in 4. Overexpression in *E. coli*, revealed a broader sequence context preference for some, including deamination at GC contexts with the DddA homologue from *S. sunii* (Ddd-Ss). Given the evidence for broader sequence preferences, the authors picked split sites based on prior work with Ddd-Bc and fused halves of the deaminase each to a TALE. The authors initially edit a ND5 mitochondrial site and compare Ddd-Bc and DDD-Ss with different split sites, where both constructs can edit, but the specific C bases changed in the editing window differ between constructs, findings replicated at a second mitochondrial site (ATP6). Across 8 broader sites average editing by the Ddd-Bc construct was ~22% while it is 28% with Ddd-Ss (Supp Table 2), with some sites accessed preferentially by one construct over the other. Focusing on several sites where GC edit might be relevant to disease, the authors compare constructs, showing that both constructs can edit in these windows, but that there is a higher rate of editing at the GC site, and that editing can be improved by adding mutations to Ddd-Ss that have been shown to improve Ddd-Bc.

Overall, this work adds a new dsDNA deaminase family member for incorporation into the genome editing toolbox. Some of the mechanistic and biochemical conclusions, such as the attribution of function to the C-terminal SPKK motif and the quantitative rigor of sequence specificity profiling, require more substantiation. More generally, the applications to genome editing suggest that the Ddd-Ss behaves differently than the Ddd-Bc construct given its different sequence preference, however it would be an overreach to say that this advance fills a substantial gap in the field or that this tool will potentiate many experiments that could not have been done otherwise. Below are a few key notes regarding precedent and impact, along with experimental and other points that would be important to address. These include:

- 1) Impact of broader exploration of dsDNA deaminase family. A major aspect of the claim towards innovation in the manuscript is the exploration of the larger dsDNA family, with preliminary characterization of eight family members and advancement of

one into genome editing. In this realm there are two precedents worth noting. As appropriately cited, recent work (Mok, 2022) has expanded the Ddd-Bc specificity beyond the initial TC to include AC and CC, thus leaving only GC as an opportunity for further expansion in this work. Second, Mougous and colleagues have looked into the broader dsDNA deaminase family (de Moraes et al, eLife, 2021) in work which was not cited here. Alternative sequence preferences were one of the findings with other family members, including an analog with activity on ssDNA, but also dsDNA activity, albeit less substantial. In the bigger picture, recent and longer standing studies on various DNA deaminases have established that each has its own sequence preference and that these can readily be altered, making the report here interesting but also somewhat incremental.

Answer: Both studies mentioned above have done beautiful work with different emphasis. We are sorry that de Moraes et al's eLife study was only cited in our methods section in previous version. We have included this citation in the introduction part after Mok's study in the revised manuscript.

2) Impact of broadening to include GC targeting. The Ddd-Ss does appear to have added the ability to target GC relative to the Ddd-Bc. A challenge with these dsDNA genome editors is that they can make multiple mutations in the same editing window, thus there is likely a very narrow set of targets for which editing of a single GC site can be achieved and is of high scientific or clinical value. In their example loci, the authors offer evidence for the fact that they have increased editing at a 'previously inaccessible' site, it was less clear that they 'reversed' a mutation specifically without local bystander edits. Perhaps by showing the CRISPResso outputs (analogous to Supp Fig S3) for the loci in Fig 2e-f it will be clear if they can edit the ND4 and ND6 loci without bystander edits to make the desired mutational change only (without bystander edits). Similarly, whether mutations introduced could be biologically valuable was not established, as there was no characterization of the editing outcomes on mitochondrial function, only sequencing based assessment.

Answer: We have added CRISPResso outputs for *ND4* and *ND6* loci in Supplementary Fig. 7 (Figure R5 below). Similar to previously reported DdCBEs, DdCBE_Ss also caused many bystander mutations. We included these data and discussion in the revised manuscript. For mitochondrial function analysis, we have measured oxygen consumption rates for cells treated with *ND4*- and *ND6*-DdCBEs. Compared to control cells treated with catalytically inactive DdCBE, cells treated with *ND6*- but not *ND4*-DdCBE_Ss5 showed lower rates of oxidative phosphorylation (Figure R6a-d). However, the lack of phenotype by *ND4*-DdCBE_Ss5 could be due to lower mtDNA editing frequencies in these cells (Figure R6e-h). These results suggest that mitochondrial mutations installed by DdCBE_Ss5 can cause biologically significant phenotypes. These data are included in Figure 3d, e and Supplementary Figure 8.

Figure R5. Allele compositions from mitochondrial DNA editing by ND4-DdCBEs and ND6-DdCBEs.

a, b, Frequencies of *MT-ND4* alleles produced by *ND4-DdCBE_Ss* (**a**) or *ND4-DdCBE_Ss5* (**b**). **c, d,** Frequencies of *MT-ND6* alleles produced by *ND6-DdCBE_Ss* (**c**) or *ND6-DdCBE_Ss5* (**d**). Shown are mean ± SD. n = 3 independent experiments.

Figure R6. Oxygen consumption rate in HEK293T cells treated with different DdCBEs.

a, b, Oxygen consumption rate (OCR) (**a**) and relative values of respiratory parameters (**b**) in sorted HEK293T cells treated with the *ND6-DdCBE_Ss5* or dead *ND6-DdCBE_Ss5*. Shown are mean \pm SD; $n = 3$ independent experiments. * $P < 0.05$ and ** $P < 0.01$ by Student's unpaired two-tailed t -test. **c, d,** Oxygen consumption rate

(OCR) (c) and relative values of respiratory parameters (d) in sorted HEK293T cells treated with the *ND4*-DdCBE_Ss5 or dead *ND4*-DdCBE_Ss5. Shown are mean \pm SD; n = 3 independent experiments. e, f, Mitochondrial base editing efficiencies of cells used in a and b (e) or cells used in c and d (f). Disease-associated target sites are shown using orange. Shown are mean \pm SD; n = 3 independent experiments. g, h, Frequencies of *MT-ND6* (g) and *MT-ND4* (h) alleles produced by *ND6*- and *ND4*-DdCBE_Ss5 in cells used in OCR measurement experiments. Shown are mean \pm SD. n = 3 independent experiments.

3) Off target activities. The dsDNA deaminase editors have raised concerns with the extent of off target activities that can occur (Lei et al, Nature 2022). Despite deaminase enzyme splitting, a significant amount of C to T transitions were observed in genomic DNA. The introduction of a new dsDNA tool comes with some obligations to similarly profile the extent of off-target effects, particularly when the construct may be more active and target different sequences. Such off-target analysis was absent in the current manuscript and would impact whether others would choose to employ these new constructs in their future work.

Answer: Thanks for the suggestion. In the revised manuscript, we have included analysis for off-target activities of DdCBE_Ss in mitochondrial and nuclear genome. We added a paragraph to describe these results (please also check Figure R1-R3 and our responses to the 2nd to 4th points of reviewer 1):

We then characterized off-target activities of Ddd_Ss. We performed assay for transposase-accessible chromatin with sequencing (ATAC-seq) to detect off-target mutations in mitochondrial genome. In all cases investigated, including *ND5.1*-DdCBE, *ATP6.1*-DdCBE and TALE-free split Ddd pairs, DdCBE_Ss induced mutations at fewer off-target sites than DdCBE_Bc (**Fig. 4a and Supplementary Fig. 9**). In addition, we checked off-target editing by *ND4*-, *ND5.1*-, and *ND6*-DdCBE in pseudogenes encoded in nuclear genome. For nuclear pseudogenes with the greatest homology (1-3 bp difference from mtDNA on-target sites), no significant off-target editing was observed (**Supplementary Fig. 10**). Recent work by Lei et al. shows that current design of DdCBEs could cause broad extent of off-target editing at non-pseudogene sites in nuclear genome. To comprehensively profile nuclear off-target editing activities of DdCBE_Ss, we performed Detect-seq experiment for HEK293T cells transfected with plasmids encoding *ND5.1*-DdCBE_Ss, *ATP6.1*-DdCBE_Ss and *ND6*-L1397-N (DdCBE_Bc from Lei et al. as a positive control). Consistent with results from Lei et al., *ND6*-L1397-N caused editing at more than 900 off-target sites in nuclear genome. *ND5.1*-DdCBE_Ss and *ATP6.1*-DdCBE_Ss caused editing at 158 and 74 off-target sites in nuclear genome (**Fig. 4b**), at a range similar to other DdCBE_Bc constructs. These data indicate that our mitochondrial DdCBE_Ss construct could induce numerous off-target editing in nuclear genome. As demonstrated by previous studies, various approaches may be applied to reduce nuclear off-target editing by DdCBE, including fusion of nuclear export signals to DdCBE, co-expression of nucleus targeted inhibitor

of DNA deaminases, or introduction of mutations to decrease the spontaneous assembly of split deaminase halves.

Other points to address:

1) The authors postulate that there is a critical role for the C-terminal SPKK motif in dsDNA recognition, attributing DNA binding activity to this region. However, an examination of the structure (PDB 6U08) reveals that the C-terminal region is on the opposite face of the enzyme from the DNA binding catalytic face. It is equally likely that the C-terminal region has some structural role or other role, and nothing to do with ‘minor groove DNA binding’. The authors should provide direct evidence for this postulated role if this conjecture is to be made.

Answer: We agree with the reviewer that we currently do not have direct evidence. During manuscript preparation and revision, we made several attempts to co-crystallize Ddd_Bc with dsDNA substrate. Unfortunately, we failed to detect any dsDNA density in the solved crystal structure. In the original manuscript, we have shown that AT-hook restored the deaminase activity of truncated Ddd_Bc. Since AT-hook has a sequence largely different from the SPKK motif of Ddd_Bc, and both AT-hook and SPKK motif have minor groove DNA binding activities, we hypothesize that SPKK motif may contribute to DddA’s function through its minor groove DNA binding activity. According to a previous study (Bharath, M. M. S., et al. Biochemistry 2002, PMID: 12056893), proline-dependent β -turn structure of SPKK motifs is important for fitting into dsDNA minor groove. When the two prolines in C-terminal SPKK motif of Ddd_Bc were mutated to valines which have similar hydrophobic side chain but can not form β -turn structure, we observed substantially reduced deamination activity by mutant Ddd_Bc. Mutating these two prolines to asparagine abolished the deamination activity of Ddd_Bc. These results support that SPKK motif may facilitate Ddd_Bc’s deamination function through its minor groove binding activity (please see **Figure R7**, also as Supplementary Figure 1b, c in the revision). However, since we do not have direct evidence, we toned down this point in the revised manuscript by saying:

“These data suggest that SPKK-related motif at the C-terminus of Ddd_Bc is important for its dsDNA deamination activity, possibly through facilitating DNA binding or other unknown structural roles.”

Figure R7. SPKK-related motifs are important for the deamination activity of Ddd_Bc.

a, Quantification of the relative amounts of deamination product versus protein concentration for Ddd_Bc (WT) and two Ddd_Bc variants (2PV, 2PN), top shows the schematic of constructs. Associated gels are shown in **b**. Shown are mean \pm SD; $n = 3$ independent experiments. **b**, In vitro cytidine deamination assays by wild type Ddd_Bc (WT) and two Ddd_Bc variants (2PV, 2PN) on 6-FAM labelled dsDNA substrate (S). The DNA sequence is 5'-FAM-ATATTATTTATTTTCATTTTATTATTATA-3'. Cytidine deamination leads to products (P) with increased mobility. Gels are representatives from $n = 3$ independent experiments.

2) Fig 1c shows quantified data and states WT data is from $n = 2$ replicates. If this is the case, error bars should likely not be shown. Also, in the legend with corresponding gels in Supp Fig S1 it states that all gels were done with $n = 3$ rather than $n = 2$. Please clarify and adjust.

Answer: WT data for 40 μ M concentration were from $n = 2$ independent experiments. We repeated experiments and now $n = 3$ for both WT and mutant Ddd_Bc. Thanks for careful reading.

3) The experimental setup in Supp Fig 2 has some issues with regard to demonstrating sequence specificity. The substrate is of a sequence 5'-FAM-N6-TCACGCC-N8'-3'. On the gel, the only thing that will appear after deamination and UDG treatment is the 5'-fragment given the 5'-FAM. Thus, if Ddd-Bc is much better at TC sites as expected, it will mask any activity at the other sequence contexts. Although the overall data, particularly with E. coli bases assays help to support the overall conclusion, this specific assay as presenting could be misleading and should be repeated with a substrate that

changes the order of the TC, AC, GC and CC sites from 5'- to 3' direction, or four substrates each of which has a different NC sequence. This will allow for more direct quantification and comparison of the enzymes.

Answer: Thanks for the suggestion. We repeated experiments using four substrates with different NC sequences. The results show that Ddd_Ss have broader sequence context compatibility than Ddd_Bc (DddA_{tox}). We included these data in Supplementary Figure 2k (Figure R8 below).

Figure R8. In vitro cytidine deamination assays by Ddd_Bc and Ddd_Ss using 6-FAM labeled dsDNA substrates with different NC contexts.

The DNA sequence is 5'-FAM-ATATTATTTGNCATTTATTATA-3', and the N is indicated at the top of each lane. Shown is a representative gel from n = 3 independent experiments.

4) Fig 1d deaminase activity column and Supp Fig 2. The data should be presented in a quantified manner. The data in Fig 1d are provided as weak, * or ** for lower, comparable versus higher activity. Implied is that this is based on a single data point from either 0.5 μ M or 10 μ M enzyme, although this is not clear. Quantitative analysis should be provided for each construct analogous to Fig 1c, given that the assay was performed in a quantitative format.

Answer: This is a great suggestion. We included quantitative analysis in the revised manuscript (Figure R9 below, also Supplementary Figure 2j).

Figure R9. Quantification of deaminase activity of different Ddd_Bc homologs.

Quantification of the relative amounts of deamination product versus protein concentration for Ddd_Bc and candidate Ddd_Bc homologs. Shown are means from n = 2 independent experiments.

5) Fig 2 and SI Table 2. In the SI Table, is the % Editing reported based on total editing across C sites in the locus of interest or based on a specific C site? As noted above, it is important to highlight whether edits are happening at specific sites or if bystander edits are common. The CRISPResso output should be provided for the sites analyzed by NGS and ideally all of the amplicons should be studied by NGS and not by Sanger estimation of editing (as Sanger does not allow for analysis of whether there are multiple edits in single loci).

Answer: In the SI Table 2, % editing for the C site with the highest editing efficiency in each targeting frame was shown. In the revised manuscript, we have replaced all Sanger sequencing results with NGS data (**Figure R10**, also Supplementary Figure 4). Accordingly, we updated the % Editing in SI Table. Finally, CRISPResso output results have been provided for all 8 sites in the revised manuscript (**Figure R11**, also Supplementary Figure 5).

Figure R10. Ddd_Ss enable efficient editing at 8 additional mitochondrial DNA sites.

a-h, mtDNA editing efficiencies of HEK293T cells treated with *TRNA-DdCBE* (**a**), *ATP8-DdCBE* (**b**), *ATP6.2-DdCBE* (**c**), *ATP6.3-DdCBE* (**d**), *ATP6.4-DdCBE* (**e**), *COX3-DdCBE* (**f**), *TRNS2-DdCBE* (**g**), *CYTB-DdCBE* (**h**). Shown are mean \pm SD; n = 3 independent experiments.

Figure R11. Allele compositions for 8 mitochondrial DNA sites edited by DdCBE_Bc or DdCBE_Ss.

a-h, Frequencies of DdCBE edited alleles produced by *TRNA-DdCBE* (**a**), *ATP8-DdCBE* (**b**), *ATP6.2-DdCBE* (**c**), *ATP6.3-DdCBE* (**d**), *ATP6.4-DdCBE* (**e**), *COX3-DdCBE* (**f**), *TRNS2-DdCBE* (**g**), *CYTB-DdCBE* (**h**). Shown are mean ± SD. n = 3 independent experiments.

6) A very minor point, but given Genus species (Capital, lowercase) nomenclature and

typical descriptions of such constructs in the field, it would seem more appropriate to refer to Ddd-Ss rather than Ddd-SS. On a related note, it may be easier to follow the experiments and comparisons if the prior Ddd is referred to as Ddd-Bc rather than DddAtox as they are both DddAtox, just from different species.

Answer: We have made these corrections as suggested in the revised manuscript. Thanks!

Reviewer #3 (Remarks to the Author):

Mi et al. present DdCBEs containing DddA_SS, a DddA homolog from *Simihoa Sunii*, enabling mitochondrial DNA editing in a previously inaccessible GC context. The authors found that the SPKK motif at the C terminus of DddA homologs are essential for deaminase activity in vitro and that DddA_SS catalyzes cytosine deamination at GC context. They used DdCBEs containing DddA_SS to install mitochondrial disease-associated mutations in human cells. This study expands mitochondrial DNA editing and provides novel therapeutic opportunities. I have the following points to improve this manuscript.

1. What is the basis of choosing eight candidates among 555 DddA homologs?

Answer: Thanks for the questions. We selected the 8 candidates according to the PSI-BLAST score (starting from the highest score) and C-terminal sequence diversity (if a group of proteins have similar C-terminal sequences, only the one with the highest score is selected for further testing). We have included this information in the methods section of revised manuscript.

2. The authors showed that the E1370N mutation in a loop region increases editing efficiency and sequence compatibility but did not explain the rationale behind this particular mutation. Was it chosen among (how) many mutants in the three loop regions?

Answer: For loop2 region, there are two different amino acids (**Figure R12**, also Supplementary Figure 11a). We mutated them individually, and found that only the E1370N mutation improves editing efficiency and sequence context compatibility of DdCBE_Bc. In the revised manuscript, we included results from the other mutation (P1369T) which reduces the activity of DdCBE_Bc (**Figure R13**, also Figure 5).

Figure R12. Multiple alignment of Ddd_Bc, Ddd_Ru, Ddd_Ss and Ddd_Fa. Secondary structure elements are presented on top (helices with squiggles, β -strands with arrows and turns with TT letters) according to Ddd_Bc structure. Loop numbers are manually designated. Two orange boxes mark two amino acids P1369 and E1370 in Ddd_Bc and corresponding amino acids in Ddd_Ss. Green boxes mark the sequences that are used to make Ddd_Bc_L1 and Ddd_Bc_L3 variants.

Figure R13. E1370N mutation extensively increases the editing efficiency and sequence compatibility of DdCBE_Bc.

a, mtDNA editing efficiencies of HEK293T cells treated with *ND5.1*-DdCBE with different Ddd_Bc variants. For Ddd_Bc_L1 and Ddd_Bc_L3, loop 1 and loop 3 sequences in Ddd_Bc are replaced with corresponding sequence of Ddd_Ss as shown in Figure R12. Shown are mean \pm SD; $n = 3$ independent experiments. *** $P < 0.001$ by Two-way ANOVA followed with Dunnett's test. **b**, **c**, mtDNA editing efficiencies of HEK293T cells treated with *ND1*-DdCBE (**b**) and *ND5.2*-DdCBE (**c**). Shown are mean \pm SD; $n = 3$ independent experiments. Ddd_Bc_E1370N was compared against Ddd_Bc. ** $P < 0.01$, *** $P < 0.001$ by Student's unpaired two-tailed t -test.

3. The authors need to profile off-target activity of DdCBE_SS in mtDNA at minimum and also in genomic DNA. It is also important to check whether DdCBE_SS is cytotoxic in comparison with DdCBEs.

Answer: In the revised manuscript, we have included analysis for off-target activities of DdCBE_Ss in mitochondrial and nuclear genome. We added a paragraph to describe these results (please also check **Figure R1-R3** and our responses to the 2nd to 4th points of reviewer 1):

We then characterized off-target activities of Ddd_Ss. We performed assay for transposase-accessible chromatin with sequencing (ATAC-seq) to detect off-target mutations in mitochondrial genome. In all cases investigated, including *ND5.1*-DdCBE, *ATP6.1*-DdCBE and TALE-free split Ddd pairs, DdCBE_Ss induced mutations at fewer off-target sites than DdCBE_Bc (**Fig. 4a and Supplementary Fig. 9**). In addition, we checked off-target editing by *ND4*-, *ND5.1*-, and *ND6*-DdCBE in pseudogenes encoded in nuclear genome. For nuclear pseudogenes with the greatest homology (1-3 bp difference from mtDNA on-target sites), no significant off-target editing was observed (**Supplementary Fig. 10**). Recent work by Lei et al. shows that current design of DdCBEs could cause broad extent of off-target editing at non-pseudogene sites in nuclear genome. To comprehensively profile nuclear off-target editing activities of DdCBE_Ss, we performed Detect-seq experiment for HEK293T cells transfected with plasmids encoding *ND5.1*-DdCBE_Ss, *ATP6.1*-DdCBE_Ss and *ND6-L1397-N* (DdCBE_Bc from Lei et al. as a positive control). Consistent with results from Lei et al., *ND6-L1397-N* caused editing at more than 900 off-target sites in nuclear genome. *ND5.1*-DdCBE_Ss and *ATP6.1*-DdCBE_Ss caused editing at 158 and 74 off-target sites in nuclear genome (**Fig. 4b**), at a range similar to other DdCBE_Bc constructs. These data indicate that our mitochondrial DdCBE_Ss construct could induce numerous off-target editing in nuclear genome. As demonstrated by previous studies, various approaches may be applied to reduce nuclear off-target editing by DdCBE, including fusion of nuclear export signals to DdCBE, co-expression of nucleus targeted inhibitor of DNA deaminases, or introduction of mutations to decrease the spontaneous assembly of split deaminase halves.

It is also important to check whether DdCBE_SS is cytotoxic in comparison with DdCBEs.

Answer: We have included cytotoxicity data in the revision (**Figure R14**, also as Supplementary Figure 6e in the manuscript). In our experiments, both DdCBE_Ss and DdCBE_Bc did not show significant cytotoxicity.

Figure R14. DdCBE_Ss has little impact on cell viability.

Cell viability was measured by recording the luminescence in HEK293T cells at various days after treated with DdCBE_Bc or DdCBE_Ss targeting different mtDNA loci. The values were normalized to the untreated samples from the same day. Shown are mean \pm SD; n = 3 independent experiments.

4. The authors need to provide full DNA sequences of DdCBE_SS including TALE sequences.

Answer: We have provided DNA sequences of DdCBE_Ss and associated TALE sequences in the revised manuscript (**Supplementary sequence 1** in supplementary information file).

5. How many cell lines were used to show the activity of DdCBE_SS. At least two different cell lines must be used.

Answer: We appreciate the reviewer's suggestion. We only tested HEK293T in the original manuscript. In the revision, we performed experiments in two additional cell lines, HeLa and U2OS (**Figure R15a-d**, also as Supplementary Fig. 6a-d). Results for HEK293T cells with the same construct are shown here for comparison (**Figure R15e, f**). These data show that DdCBE_Ss can work well in other cell lines.

Figure R15. DdCBE_Ss can enable mitochondrial base editing in three different human cell lines.

a, b, mtDNA editing efficiencies of HeLa cells treated with *ND5.1*-DdCBE (**a**) and *ATP6.1*-DdCBE (**b**). **c, d**, mtDNA editing efficiencies of U2OS cells treated with *ND5.1*-DdCBE (**c**) and *ATP6.1*-DdCBE (**d**). **e, f**, mtDNA editing efficiencies of HEK293T cells treated with *ND5.1*-DdCBE (**e**) and *ATP6.1*-DdCBE (**f**). Shown are mean \pm SD; n = 3 independent experiments.

6. Most of Supplementary Figures must be shown as main figures. There are only two main figures.

Answer: We have added new data and reorganized figures as suggested. The revised manuscript now includes 5 main figures and 11 supplementary figures. Thank you for the advice.

Reviewers' Comments:

Reviewer #1:

Remarks to the Author:

The authors have done a considerable amount of revision, and added important sequencing information on off-target potential for these mutations. I have no further comments on the manuscript and suggest publication.

Reviewer #2:

Remarks to the Author:

This revised manuscript by Mi, Shi, Wang and colleagues explores the Ddd-Ss homolog as a dsDNA deaminase that can be applied in mitochondrial base editing in an expanded sequence context.

On the initial review, my summary assessment was that additional rigor was necessary to establish aspects of the conclusions (more below), but additionally that "it would be an overreach to say that this advance fills a substantial gap in the field or that this tool will potentiate many experiments that could not have been done otherwise". Overall, in this revision the authors have done an outstanding job of addressing the experimental requests, although these additions notably do not change my opinion of the anticipated modest impact of the work. Nonetheless, as impact/importance questions are separate from questions of experimental rigor, it is worth highlighting progress on the front of experimental rigor made in this revision.

Specifically, prior concerns that have been addressed include:

(1) Prior exploration of the broader dsDNA deaminase family. The precedents are now better cited, which is appreciated. The citations, however, only in part address the statement in my prior review holds that "recent and longer standing studies on various DNA deaminases have established that each has its own sequence preference and that these can readily be altered, making the report here interesting but also somewhat incremental."

(2) Off target activities. The authors have done an outstanding job of now quantifying various forms of off-target activity. These newly incorporated experiments do not suggest any 'advantage' of their new editors with regards to off target activities, but are nonetheless very helpful for characterizing their modified editors. Off-target activity remains a major limitation of this class of editors.

(3) The revised manuscript has appropriately now offered more cautious interpretation of the SPKK motif. The authors may want to reflect on pre-print structure of the DddA-dsDNA complex, which on first glance (<https://doi.org/10.21203/rs.3.rs-2031914/v1>, PDB not available) does not appear to me to suggest that 'minor groove binding' by the C-terminus, which they had initially proposed, will be very likely.

(4) Other changes were appreciated and appear to add rigor (added replicates, improved biochemical assay isolating each sequence context, presentation of all data output with CRISPResso, and nomenclature change from DDD-SS to DDD-Ss).

Reviewer #3:

Remarks to the Author:

The authors have addressed my concerns in their revised manuscript.

We thank all the reviewers for their efforts and critical thoughts during the reviewing process. We are happy to know that all reviewers are satisfied with the revised version. In particular, we thank the reviewer #2 for pointing out potential limitations of our study. We continue working on improving our DdCBEs and expanding their applications. Hopefully we will have more important progress to report in future. Our point-to-point responses are listed below in light blue text.

REVIEWERS' COMMENTS

Reviewer #1 (Remarks to the Author):

The authors have done a considerable amount of revision, and added important sequencing information on off-target potential for these mutations. I have no further comments on the manuscript and suggest publication.

Answer: Thank you!

Reviewer #2 (Remarks to the Author):

This revised manuscript by Mi, Shi, Wang and colleagues explores the Ddd-Ss homolog as a dsDNA deaminase that can be applied in mitochondrial base editing in an expanded sequence context.

On the initial review, my summary assessment was that additional rigor was necessary to establish aspects of the conclusions (more below), but additionally that “it would be an overreach to say that this advance fills a substantial gap in the field or that this tool will potentiate many experiments that could not have been done otherwise”. Overall, in this revision the authors have done an outstanding job of addressing the experimental requests, although these additions notably do not change my opinion of the anticipated modest impact of the work. Nonetheless, as impact/importance questions are separate from questions of experimental rigor, it is worth highlight progress on the front of experimental rigor made in this revision.

Answer: Thank the reviewer #2 for positive comments on our experimental improvements.

Specifically, prior concerns that have been addressed include:

(1) Prior exploration of the broader dsDNA deaminase family. The precedents are now better cited, which is appreciated. The citations, however, only in part address the statement in my prior review holds that “recent and longer standing studies on various DNA deaminases have established that each has its own sequence preference and that these can readily be altered, making the report here interesting but also somewhat incremental.”

Answer: We agree with the reviewer that based on previous experiences, DNA deaminases from different species may have different sequence preferences that can be altered though protein engineering or evolution. That is the reason why we have looked for dsDNA deaminases in different species. Luckily, we identified a dsDNA deaminase with high activity towards GC context and successfully converted it to a mitochondrial base editor that can access previously inaccessible sites.

(2) Off target activities. The authors have done an outstanding job of now quantifying various forms of off-target activity. These newly incorporated experiments do not suggest any ‘advantage’ of their

new editors with regards to off target activities, but are nonetheless very helpful for characterizing their modified editors. Off-target activity remains a major limitation of this class of editors.

Answer: Although in many cases, our new editors show fewer off-target editing in both mitochondrial and nuclear genome, we agree with the reviewer that off-target activity remains a major limitation of current mitochondrial base editors. Protein evolution or other novel approaches are needed to address this concern. Future work in our lab will hopefully help solve this limitation.

(3) The revised manuscript has appropriately now offered more cautious interpretation of the SPKK motif. The authors may want to reflect on pre-print structure of the DddA-dsDNA complex, which on first glance (<https://doi.org/10.21203/rs.3.rs-2031914/v1>, PDB not available) does not appear to me to suggest that 'minor groove binding' by the C-terminus, which they had initially proposed, will be very likely.

Answer: Thanks for the information. The DddA-dsDNA complex structure in this preprint used the toxin domain (Gly1290 to Pro1422). The full length of toxin domain contains Gly1290 to Cys1427. As a result, their structure only contains the first SPKK-related motif (Ser1418 to Lys1420). We agree with the reviewer that there is no evidence supporting the direct role of C-terminus in the minor groove binding. Interestingly, we noticed that "Binding of DddA to the bent DNA is also supported by interaction with the backbone phosphate groups from both strands, involving residues Ser1331, Asn1339, Tyr1340, Lys1402, and Lys1420 (Supplementary Fig. 2)". Therefore, the structure data support our interpretation in the manuscript that "SPKK-related motif at the C-terminus of Ddd_Bc is important for its dsDNA deamination activity, possibly through facilitating DNA binding or other unknown structural roles". We appreciate the reviewer's careful analysis. We hope that the structure of full length toxin domain DddA-dsDNA complex will eventually reveal the molecular function of the C-terminus.

(4) Other changes were appreciated and appear to add rigor (added replicates, improved biochemical assay isolating each sequence context, presentation of all data output with CRISPResso, and nomenclature change from DDD-SS to DDD-Ss).

Answer: We appreciate the reviewer's positive comments for praising our efforts to improve the manuscript. It would be impossible without your precious suggestions.

Reviewer #3 (Remarks to the Author):

The authors have addressed my concerns in their revised manuscript.

Answer: Thank you.